# Surveillance of Amoebic Keratitis-Causing Acanthamoebae for Potential Bacterial Endosymbionts in Ontario, Canada

**DOI:** 10.3390/pathogens11060661

**Published:** 2022-06-08

**Authors:** Nessika Karsenti, Andrew Purssell, Rachel Lau, Filip Ralevski, Shveta Bhasker, Hira Raheel, Andrea K. Boggild

**Affiliations:** 1Schulich School of Medicine & Dentistry, London, ON N6A 5C1, Canada; nkarsenti2023@meds.uwo.ca (N.K.); hira.raheel@mail.utoronto.ca (H.R.); 2Faculty of Medicine, University of Ottawa, Ottawa, ON K1N 6N5, Canada; apurssell@toh.ca; 3Public Health Ontario Laboratories, Public Health Ontario, Toronto, ON M5G 1M1, Canada; rachel.lau@oahpp.ca (R.L.); filip.ralevski@oahpp.ca (F.R.); 4Institute of Health Policy, Management and Evaluation, Dalla Lana School of Public Health, University of Toronto, Toronto, ON M5S 1A1, Canada; shveta.bhasker@mail.utoronto.ca; 5Tropical Disease Unit, Division of Infectious Diseases, UHN-Toronto General Hospital, Toronto, ON M5G 2C4, Canada; 6Department of Medicine, University of Toronto, Toronto, ON M5S 1A1, Canada; 7Institute of Medical Science, University of Toronto, Toronto, ON M5S 1A1, Canada

**Keywords:** surveillance, *Acanthamoeba*, endosymbionts, amoebic keratitis

## Abstract

*Acanthamoeba* spp. are the causative pathogens of several infections, including amoebic keratitis (AK), a vision-threatening infection. Acanthamoebae from corneal specimens of patients with AK harbor bacterial endosymbionts, which may increase virulence. We sought to understand the spectrum of bacterial endosymbionts present in clinical isolates of *Acanthamoeba* spp. identified in our reference parasitology laboratory. Isolates of *Acanthamoeba* spp. obtained from our biobank of anonymized corneal scrapings were screened for potential endosymbionts by PCR using primer pairs detecting bacteria belonging to orders Chlamydiales, Rickettsiales, or Legionellales and pan16S primers. Three primer pairs specific to the 18s rRNA gene of *Acanthamoeba* spp. were used for the amplification of *Acanthamoeba* DNA used for sequencing. Sanger sequencing of all PCR products was performed, followed by BLAST analysis for species identification. We screened 26 clinical isolates of *Acanthamoeba* spp. for potential endosymbionts. Five isolates (19%) were found to contain bacterial DNA belonging to Legionellales. Three (11%) contained members of the Rickettsiales and *Pseudomonas genticulata* was detected in a *Rickettsia*-positive sample. One strain (4%) contained *Neochlamydia hartmannellae*, a member of the Chlamydiales order. Bacterial endosymbionts are prevalent in clinical strains of *Acanthamoeba* causing AK isolated from corneal scrapings. The demonstration of these organisms in clinical *Acanthamoeba* isolates supports a potential exploration of anti-endosymbiont therapeutics as an adjuvant therapy in the treatment of AK.

## 1. Introduction

*Acanthamoeba* spp. are ubiquitous, free-living amoebae, which, when present in the eye, are capable of causing amoebic keratitis (AK), especially in the context of poor contact lens hygiene [1]. This disease manifests clinically with corneal ulceration, which can progress to blindness even with prolonged courses of anti-acanthamoebal therapy [1].

*Acanthamoeba* spp. subsist off bacteria found in their environment. However, some bacteria can avoid digestion and persist within the amoeba as an endosymbiont. Stable endosymbiosis has been described for several Proteobacteria and Chlamydiae [2,3,4,5,6]. Bacteria belonging to the Legionellales family have been found to successfully replicate within *Acanthamoeba* spp. [7]. Cytopathogenicity by *Acanthamoeba* spp. increases in vitro on fibroblast monolayers when bacterial endosymbionts related to Legionellales or Chlamydiales are present [8,9,10]. The presence of antimicrobials directed against endosymbionts during *Acanthamoeba* infection in a 3D corneal tissue model attenuated inflammatory markers and cytopathic effects [10]; as such, it is possible that the presence of endosymbionts in vivo increases disease severity in AK.

A recent study has demonstrated an outbreak of AK in the UK, where the incidence has been rising drastically since 2011 [11], and this is occurring independently of the changes to the number of contact lens wearers or diagnostic method [11]. Previous studies have estimated the presence of intracellular bacteria found in either clinical isolates or environmental isolates of *Acanthamoeba* to be between 24% and 59% [8,9]. This range of prevalence is partially attributable to differing detection techniques and the difficulties of detecting a wide range of endosymbiont species. In this study, we provide epidemiological data regarding AK in the province of Ontario, Canada. We also provide data on bacterial endosymbiont prevalence through the screening of clinical samples using both broad and order-specific bacterial PCR primers. Our aims are to: (i) determine the rate of detection of *Acanthamoeba* in patients presenting with keratitis; (ii) identify the *Acanthamoeba* species responsible; and (iii) determine if the detection of an endosymbiont was associated with age or sex. We attempt to draw conclusions regarding the *Acanthamoeba* spp. responsible for harboring various endosymbionts in clinical samples in Ontario. Categorizing the prevalence of bacterial endosymbiosis in clinical AK infections may help us to infer how endosymbionts potentially contribute to virulence, which may support the development of more effective and targeted treatments.

## 2. Results

**Patient characteristics and incidence of *Acanthamoeba***: Between 2009 and 2016, a total of 739 samples corresponding to 642 individual patients were tested for *Acanthamoeba* by our province’s clinical parasitology lab. Of these, 86 (11.6%) samples corresponding to 50 (7.8%) individual patients were positive (Table 1). Of the 642 individuals who submitted sample(s) for testing for AK, 332/642 (51.7%) were female, 306/642 (47.7%) were male and 4/642 (0.6%) were of unknown sex. Mean age was 50.4 years old (standard deviation 22.74) with a median of 51 (ranging from 4 days to 94 years old). The majority of samples submitted were corneal scrapings (676/739, 91.5%); other sample types submitted were contact lens solutions (47/739, 6.4%) or culture plates (1/739, 0.1%); 15 sample submissions were of unrecorded sample types (15/739, 2.0%) (Table 1).

**Yearly infection rates:** We found an increasing trend of infection rates characterized by a sharp rise in 2015 that almost doubled the number of positive AK infections detected in clinical samples by our reference parasitology laboratory. Linear regression analysis for infection rates between 2009 and 2016 showed that there was a yearly increase of 2.19 cases per 100 patient submissions (95% Confidence Interval (CI) 0.71 to 3.69, *p* = 0.001) (Table 2). Of note, in 2015, our reference laboratory introduced an additional *Acanthamoeba* molecular test, which may have increased sensitivity and contributed to the increased rate of detection [12]. A separate analysis was performed for 2009–2014, showing a yearly increase of 1.21 cases per 100 patient submissions (95% CI −0.13 to 2.54, *p* = 0.066) (Figure 1). Analysis of infection proportion per month revealed no seasonal pattern to AK infections (Figure 2).

**Association of AK, causative species, and endosymbiont detection with age and sex (Table 3, Table 4 and Table 5****):** There were 50 positive cases throughout the study period. Positivity of AK in males was 6.2% (19/306) and in females was 9.0% (30/332) (one unknown), with an odds ratio of males over females of 0.66 (95% CI 0.36 to 1.19, *p* = 0.1803) (Table 3). The mean age of infection was 44.5 +/− 20.26 years old (median 46 years, range 4 years–92 years) compared to those without infection of 51.1 +/− 22.97 years old (median 53 years, range 4 days–94 years), with a mean difference of 6.63 ± 3.35 (95% CI 0.05 to 13.22, *p* = 0.048). No association of AK with sex was observed (Table 3); however, there was a correlation between a younger patient age and positivity of the sample (mean difference 6.63, 95% CI 0.051 to 13.22, *p* = 0.0482). 

**Identification of *Acanthamoeba* species and detection of endosymbionts**: Of the 50 qualifying positive individual cases, only 26 *Acanthamoeba*-positive samples from 25 different patients (49%) had sufficient DNA remaining for *Acanthamoeba* species identification and analysis of endosymbionts. Samples A22 and A23 were obtained from the same patient. Information on both samples is included in Table 6 since both samples were used in subsequent experiments for confirming endosymbiosis. Another sample, A4, was also included in the initial dataset but later removed from analysis due to our exclusion criteria of abscess aspirates sample type (not shown in Table 6). To determine if there was a selection bias on the subset with DNA available for analysis for endosymbionts, the proportion of sex and mean age of those samples available for analysis versus non-available was analyzed. Sex of patients with a sufficient sample for analysis was 64% female versus 46% female without sufficient material (*p* = 0.773), and a mean age of 43.4 +/− 22.80 (median: 46 years, range 2 years–80 years) vs. 44 years old (*p* = 0.811), respectively. Therefore, there was no discernible selection bias on the small subset of samples available for the identification of *Acanthamoeba* species and endosymbiont analysis. 

Only in roughly half of the samples (12 of 25) could the *Acanthamoeba* be identified at the species level. These included *A. castellanii* (4/12, 33%), *polyphaga* (2/12, 17%), *A. hatchetti* (2/12, 17%), *A. quina* (1/12, 8%), *A. lenticulata* (2/12, 17%), and *A. griffini* (1/12, 7%) (Table 6). A summary of each species’ corresponding genotype is presented in Table 7. Comparing the species according to sex or age within positive samples showed no association (Table 4). The proportion of *A. castellanii* species in males was 16.7% (1/6) and in females was 50% (3/6), with the odds ratio of having *A. castellanii* in males to females of 0.2 (95% CI 0.013 to 2.18, *p* = 0.5455) (Table 4). Mean age of patients with *A. castellanii* was 39.5 +/− 24.09 (median: 38.5 years, range: 12 years–69 years), and of those with other species was 48.6 +/− 15.27 (median: 47 years, range 23 years–70 years). Mean age difference was −9.125 (95% CI −15.93 to 34.18 *p* = 0.4360). 

Of the 25 unique clinical specimens, potential endosymbionts were found in 8/25 (32.0%) (Figure 3 and Table 6). Of these eight, five (62.5%) were of the order Legionallales, two (25%) were of the order Rickettsiales, and one (12.5%) was a Chlamydiales. *Pseudomonas genticulata* was also detected in one of the Rickettsiales-positive samples. A possible co-endosymbiont of Bacillaciae in a Legionellales-positive sample was also detected (sample A14), with a low 84% identity in the BLAST search (Table 6). 

No associations of sex or age were detected when comparing *Acanthamoeba*-positive samples that contained endosymbionts to those that did not. The number of male samples with endosymbionts was 4/9 (44.4%) and in females was 4/12 (33.3%), with an odds ratio of 2.4 (95% CI 0.48 to 12.63) *p* = 0.3942. Using Fisher’s exact test (Table 5), the mean age in endosymbiont-containing samples was 42.1 +/− 24.70 (median: 46, range: 4 years–69 years) compared to 45.9 +/− 22.54 (median: 46 years, range: 2 years–80 years) in non-endosymbiont-containing samples, with a mean difference of 3.82 (95% CI −16.78 to 24.41), *p* = 0.705, using the T-test (Table 5). All of the Legionellales detected came from corneal scrapings. Two (66%) Rickettsiales samples came from contact lenses in saline, with the third one being of unknown origin. The Rickettsiales-positive sample in which *Pseudomonas genticulata* was found as well was from an unrecorded sample type. The Chlamydiales-containing sample was detected from a contact lens in saline. Of the nine detected potential endosymbionts, only the Chlamydiales was successfully identified to the species level, revealing *Neochlamydiae hartmanellae*. We explored associations between *Acanthamoeba* spp. and endosymbionts detected (Table 6). Four of the eight endosymbiont-positive samples from different patients could be correlated with *Acanthamoeba* species. Of these, two samples containing Legionellales were determined to be in *A. polyphaga*, whereas the three remaining samples were in unknown *Acanthamoeba* spp. The two Rickettsiales-containing samples were species identified to be in *A. castellanii*. The *Acanthamoeba* spp. containing Chlamydiales was not identified. Other species identified, including *A. griffini*, *A. hatchetti*, *A. lenticulate*, and *A. quina*, did not contain endosymbionts. Statistical analysis was not attempted to correlate endosymbionts and *Acanthamoeba* spp. due to the small sample size. 

**Confirming endosymbiosis**: Three samples (A5, A22, and A23) were further examined to eliminate the possibility that the detected bacteria existed outside of *Acanthamoeba* cysts. We were unable to confirm this with other samples due to an insufficiency in sample material. No Legionellales DNA was detected outside of cells when omitting the freeze–thaw step, and only one of the Rickettsiales samples (A22) showed extracellular DNA (Figure 4). 

## 3. Discussion

Amoebic keratitis caused by *Acanthamoeba* spp. can develop in immunocompetent patients through exposure to contaminated water or improper contact lens use. This disease causes corneal ulcers that are notoriously difficult to treat and often result in the need for a corneal transplant [1]. Understanding the profile of AK infections through surveillance efforts becomes necessary for informing treatment. Furthermore, *Acanthamoeba* infections have been shown to be more virulent when an endosymbiont is present [10]. Understanding the extent of these relationships is also important for determining the optimal treatment—specifically, the addition of endosymbiont-directed therapy may attenuate the severity of infection. Through our work, we have provided information on the state of AK infections in Ontario, shown reason for monitoring the potential increase in the prevalence of AK in Ontario, and described the endosymbiotic relationships between bacteria and parasite in *Acanthamoeba* spp. causing keratitis.

No correlation between sex and *Acanthamoeba* infection has been found in our study population, encompassing all patients in Ontario who have submitted samples for query *Acanthamoeba* testing from July 2009 to July 2016. However, positivity tends to cluster with younger patients; this is consistent with evidence showing younger age as a predictor of complications arising from contact lens use [14]. 

Between 2009 and 2016, an average yearly doubling of the infection rate was seen, but this is likely attributed to a change in the diagnostic strategy to increase sensitivity in 2015. To avoid this bias, analysis was restricted to 2009–2014. Although not statistically significant, there was a trend towards a yearly increase of 21%. It was found that swimming in public pools or hot tubs, insufficient hand washing, and showering while wearing contact lenses were risk factors for acquiring AK during the outbreak in the UK. Patients in Ontario may also be practicing these behaviors [11].

Infections in Ontario have been found to be caused by a variety of species, including *A. castellanii*, *A. polyphaga*, *A. hatchetti*, *A. quina*, *A. lenticulata*, or *A. griffini*. It has been shown that *A. lenticulata*, *A. hatchetti*, and *A. castellanii* are among the most pathogenic species [15], with *A. castellanii* representing the most frequent infections. This is consistent with our findings—the most abundant species detected was *A. castellanii*, making up 5 of the 12 sequenced infections. There was no correlation between age or sex and the species of *Acanthamoeba* found in the clinical sample. It is possible that species of *Acanthamoeba* other than the ones listed here were responsible for the disease in patients. Furthermore, the PCR primers used for the sequencing of the genetic material have been known to have low sensitivity [16,17], which could account for the samples that we were unable to identify. Moreover, species identification is dependent upon sequences deposited in the NCBI database; therefore, a new or underrepresented species would have been missed. 

Endosymbiotic bacteria have been found to enhance virulence for in vitro studies, including simulated corneal models [10]; surveying clinical samples for the presence of various endosymbionts can provide information on the prevalence of these endosymbiotic relationships and therefore aid in determining appropriate treatment strategies. No evidence for a correlation between the sex or age of patients and the presence of endosymbionts was found. Of 26 clinical samples surveyed, two from the same patient, endosymbionts belonging to the Chlamydiales, Rickettsiales, or Legionellales bacterial orders have been detected in nine, two of which were from the same patient (33%). The Chlamydiales isolate was the only one that could be identified to the species level and was determined to be *Neochlamydiae hartmanellae*. This organism has been previously found to be an endosymbiont of the amoeba *Hartmanella vermiformis* [15], another free-living amoeba that has also been shown to host members of the Legionellales order [18]. It is therefore likely that these amoebas form endosymbiotic relationships with the same mechanisms, and this points towards free-living amoeba as a potential reservoir for Chlamydiales proliferation, similar to the well-characterized relationship between *Acanthamoeba* and *Legionella pneumophilia* [7]. 

Fritsche and colleagues [19] reported an endosymbiont prevalence of 24% within their samples. Our rate of potential endosymbiont detection within our samples was higher at 32%, but this may be due to improved sensitivity with molecular methods compared to electron microscopy. Iovieno and colleagues [9] included mycobacteria and *Pseudomonas* in their surveillance and reported a 59.4% occurrence, which is likely an underrepresentation as these were species-specific PCRs. Our pan-bacterial 16S PCR has allowed for the detection of a *Pseudomonas* endosymbiont, but we have not detected any mycobacteria in our samples. Since *Acanthamoeba* spp. from clinical samples have been found to harbor mycobacterial endosymbionts [9] and a *Mycobacterium* spp. endosymbiont was found in an ATCC strain collected from a human cornea [19], the absence of mycobacteria in our study could be due to the small sample size. The detection of a *Mycobacterium* spp. endosymbiont in a clinical sample detected at our reference laboratory after 2016 (data not shown) supports this supposition. Alternative explanations include geographic restriction due to climatological and ecosystemic factors. 

Species identification of the *Acanthamoeba* spp.-containing endosymbionts has revealed that the Rickettsiales detected in our study were found in only *A. castellanii*, and Legionellales was always found in *A. polyphaga* (two of five samples positive for Legionellales; *Acanthamoeba* species could not be identified in the three other samples). No endosymbionts were found in species other than *A. castellanii* or *A. polyphaga*, as the former is one of the more virulent species [20] and its ability to harbor endosymbionts is of importance. This suggests that certain pairings of *Acanthamoeba* spp. and endosymbionts could yield more stable endosymbiosis, contribute to increased virulence, or convey a superior fitness advantage to the organisms. However, we had a small dataset and further studies with large sample sizes are warranted. Determining the genotype of represented species was limited by the quantity of the specimen available; however, *A. castellanii* has been shown to correspond to genotypes T1 or T4, and *A. polyphaga* to T2 or T4 [13] (see Table 7). It has been previously shown that the T4 genotype is the most pathogenic [21]. Correlating genotype with the presence of endosymbionts could be an interesting avenue of further study. 

It is interesting to note that although Chlamydiales are common endosymbionts in environmental *Acanthamoeba* samples [7], we only detected one in our clinical samples. The lower proportion of Chlamydiales detected could reveal the lower affinity of this family for *Acanthamoeba* species that cause AK or a decreased ability to enhance *Acanthamoeba* virulence. The suspected Bacillaciae species was identified but had low sequence homology. It is therefore possible it is not truly Bacillaciae or may be a novel species. Further sequencing efforts to determine its identity were unsuccessful. 

To determine if the bacteria detected in clinical samples were present within the *Acanthamoeba* cells, thereby potentially increasing virulence, DNA extraction was repeated without using serial freeze–thaws, which would lyse *Acanthamoeba* cells. An absence of DNA would therefore indicate if the bacteria originally detected were located inside the cell, versus simply being present as a contaminant in the sample environment at the time of collection. While this does not confirm endosymbiosis, it provides evidence that the detected bacteria were present within *Acanthamoeba* and are potential endosymbionts. Extraction without freeze–thaw was done on three of the nine samples containing potential endosymbionts; two Legionellales- and one Rickettsiales-containing sample. No Legionellales DNA was detected with this procedure, supporting the presence of the Legionellales endosymbiotic in this sample. The endosymbiotic relationship between Legionellales and *Acanthamoeba* is well characterized; we conclude that this sample is therefore most likely a true case of endosymbiosis. Due to the abundance of literature on the relationship between *Acanthamoeba* spp. and Legionellales endosymbionts, it is likely that the other four detected Legionellales samples represent true endosymbionts. Of two Rickettsiales samples tested, one showed the presence of extra-amoebal Rickettsiales DNA, whereas the other one did not. Rickettsiales and *Acanthamoeba* co-infections may therefore be the result of a true endosymbiotic relationship, or simply a mixed infection. This protocol could not be carried out on the Chlamydiales-containing isolate due to an absence of material, but the literature has shown this relationship in other samples [6,10]. An instance of *Pseudomonas genticulata* was found in a sample also positive for Rickettsiales—there is evidence of this species being a true endosymbiont [9]. Other papers have examined endosymbiosis within samples using electron microscopy [3,4,5,6,9]. However, because this study was retrospective in nature, primary clinical samples were not available for microscopy use. 

Due to difficulty in obtaining species identifications for *Acanthamoeba* spp., and the paucity of sample material available for further study after clinical diagnosis has been established, our results only represent a small dataset. Connections between specific *Acanthamoeba* species and endosymbiont species should be taken cautiously. Indeed, molecular-based identification of acanthamoebae may be hindered by sequence heterogeneity across species, necessitating the use of a composite reference standard of combinations of primer sets applied in a stepwise or algorithmic manner to ensure accurate detection [12]. It is also possible that the fraction of samples that we found to contain endosymbionts is an underestimation because we did not actively seek out *Pseudomonas* or mycobacterial endosymbionts, as other studies have done [9]. 

The frequent detection of *Pseudomonas* endosymbionts by other research teams points to a need for higher surveillance of this bacterium in clinical samples. Iovieno and colleagues [9] found a higher prevalence of *Pseudomonas* than any other bacteria in their samples (13 of 38 isolates contained a *Pseudomonas* endosymbiont closely related to *P. aeruginosa*). Additionally, mycobacteria have been shown to infect the corneas alone [22], and *Acanthamoeba* have been shown to harbor mycobacteria. Purssell and colleagues found a *Mycobacterium* in *A. polyphaga* [10], and a mycobacterial species was found in a clinical sample sent to our reference parasitology laboratory in 2016, beyond the range of this study period (unpublished data). Surveying for endosymbiotic relationships between this bacterium and *Acanthamoeba* spp. will therefore provide information valuable for informing treatment, and harnessing the power of next-generation sequencing for endosymbionts will enable a more granular assessment of such relationships.

This study describes the profile of *Acanthamoeba* species that cause AK in Ontario, both in the prevalence of infections and state of potential endosymbiotic relationships. We have shown potential for an increase in the incidence of AK in Ontario, and that endosymbiotic relationships with bacteria exist with parasites causing these infections. Infection rates of *Acanthamoeba* have been shown to increase and these infections worsen in the presence of endosymbionts. Studies such as ours can provide much-needed insight into the pattern of disease and the resulting treatment of AK. 

## 4. Materials and Methods

**Epidemiological information:** Basic demographic information (age, sex) on samples sent to the Public Health Ontario reference parasitology laboratory (PHOL) for *Acanthamoeba* spp. testing from July 2009 to July 2016 was obtained from our biobank of surplus specimens. Given that the PHOL is largely responsible for testing all *Acanthamoeba* samples, the majority of clinical samples collected in Ontario were sent to our laboratory for testing. Assessing samples sent to PHOL is therefore a reliable means of assessing the burden of *Acanthamoeba* across the entire province. 

Samples were received in the form of corneal scrapings, contact lens solutions, culture plates, or abscess aspirates. As abscess aspirates were from internal organs rather than the cornea, they were not causative for *Acanthamoeba* keratitis, and thus were excluded from our dataset. No other exclusion criteria were employed for data acquisition. Data were grouped by calendar year.

Samples were anonymized and those diagnosed as positive *Acanthamoeba* infection were obtained for subsequent species identification and endosymbiont analysis. The work was classified as laboratory surveillance, and not human subject research requiring human subject considerations.

**Sequencing:** A primer set was designed for the sequencing of *Acanthamoeba* species, as previously described (Appendix A
Table A1 [23]). This was achieved by aligning 66 different sequences comprising 13 species (Appendix B
Table A2) of the 18s rRNA gene of *Acanthamoeba*, obtained from NCBI. Sequence alignments were done on Mega 6 [24] and two regions with the highest conserved homology and longest amplicon possible were selected to design the forward and reverse primers. Primer Express (Applied Biosystems, Foster City, CA, USA) was used for the design of the primers, and then they were subjected to a BLAST search to ensure no cross-reactivity with human and other corneal microbes or eukaryotic pathogens. The resulting sequencing primers were AcantSeqFwd 5′-CCTACCATGGTCGTAACGGG-3′ and AcantSeqRev 5′AGGGCAGGGACGTAATCAAC-3′ with an amplicon size of 1688bp [23]. 

**DNA extraction**: The QiaAmp DNA Mini extraction kit (Cat#50304, Qiagen, Hilden, Germany) was used with a modification to the manufacturer’s protocol. Briefly, 200 µL of clinical sample was subjected to three rounds of freezing in liquid nitrogen and thawing at 56 °C; then, 20 μL of Proteinase K and 200 μL of Buffer AL were added and incubated at 56 °C for a further 10 min. Following this, 200 μL of 95% ethanol was added to the sample, vortexed, and loaded onto a spin column for washing according to the manufacturer’s protocol. DNA was eluted in 60 μL of AE buffer. 

***Acanthamoeba*****spp. species identification:** Species identification was performed with the use of PCR amplification and Sanger sequencing. Three different end-point primer sets—Nelson primers [16], JDP primers [17], and a novel primer set designed specifically to aid in sequencing the *Acanthamoeba* 18s rRNA gene [23] (see above, “Designing novel primer set for sequencing”)—were used. PCR for the Nelson and JDP primer sets was performed using Amplitaq Gold Fast Mix (Cat# 4390939 Thermo Fischer Scientific, Waltham, MA, USA) on a Veriti 96-well fast thermal cycler (Applied Biosystems, Foster City, CA, USA), with initial denaturing at 95 °C for 10 min, and then subjected to 45 cycles of 96 °C for 5 s, 57 °C for 5 s, and 68 °C for 15 s, followed by a final extension at 72 °C for 30 s (see Appendix A
Table A1 for annealing temperatures of various primer sets used in this study). PCR for the novel primer set was carried out using 22.5 μL Accuprime Pfx mix on a Veriti 96-well fast thermal cycler (Applied Biosystems, Foster City, CA, USA) and 300 µM of each of the primers. Cycling conditions were as follows: 95 °C for 10 min followed by 45 cycles of 96 °C for 5 s, 58 °C for 5 s, and 68 °C for 1 min 45 s, with a final extension at 72 °C for 30 s. Sanger sequencing was performed on this amplified product using BigDye v3.1 and a 3130xl genetic analyzer (Applied Biosystems, Foster City, CA, USA) [23]. Sequenced fragments obtained from all primer sets for *Acanthamoeba* species identification were aligned using Contig Express (Life Technologies, Carlsbad, CA, USA). DNA sequences from NCBI BLAST that were clearly different between *Acanthamoeba* species were used for species identification. The 23 genotypes that map to putative specific epithets represented in the conventional scientific literature are summarized in Table 5 [13,21]. For the purposes of this study, historic and conventional *Acanthamoeba* species nomenclature is reported.

**Endosymbiont detection and bacterial species identification**: Detection of potential endosymbionts was carried out using 4 different primer sets. Leg225 and LEG858 [25], Rp887p and Rp1258r [26], as well as EHR16SR and EHR16SD [27] were used to detect members of *Legionellales*, *Rickettsiales*, and *Chlamydiales*, respectively. In particular, 8FPL and 806R [28] are pan-16S primers that were used for the broad detection of other bacteria. All primer sets were specific to the 16S region, except for Rp877p and Rp1258r, which targeted a citrate synthase gene [26]. PCR was carried out using the Amplitaq Gold Fast Mix (Thermo Fischer, Waltham, MA, USA) on a Veriti 96-well fast thermal cycler (Applied Biosystems, Foster City, CA, USA). They were denatured at 95 °C for 10 min, and then subjected to 45 cycles of 96 °C for 5 s, 5 s at annealing temperatures specific for each primer set (Appendix A), and 68 °C for 30 s, with final extension at 72 °C for 30 s. Amplified DNA was visualized on a 1% agarose gel containing ethidium bromide. *Rickettsia*-positive control DNA was obtained from Phoenix Airmid (Oakville, ON, Canada); *Legionella* and *Chlamydia* control DNA were obtained from our PHO laboratories. Sequencing was performed, and obtained sequences were subjected to a BLAST search to identify bacterial species with the highest sequence homology. For bacterial species where sequencing was not successful in the first PCR, the PCR product was re-amplified by excising the band from the gel and purifying it using the QIAquick Gel Extraction Kit (Cat# 28704, Qiagen, Hilden, Germany) according to the manufacturer’s protocol; it was then subjected to a second round of PCR and Sanger sequencing. 

**Confirmation of endosymbiosis**: To confirm that bacteria detected from *Acanthamoeba* samples were present within *Acanthamoeba* cells and not from ocular flora or external sources, a subset of potentially endosymbiont-positive *Acanthamoeba* primary samples were subjected to DNA extraction without the serial freeze–thaw step. This step is required for the effective lysis of *Acanthamoeba* cysts; therefore, comparing this fraction to the one where freeze–thaw was conducted could help to determine whether the bacterium was an endosymbiont of *Acanthamoeba*. 

**Statistical analysis:** Annual infection rates were calculated by dividing the number of patients positive for AK by the total number of patients whose samples were sent for testing that year (Figure 1 and Table 2). Yearly infection rates and data on age or sex were analyzed using Student t-test, Chi-square tests, or Fisher’s exact test as appropriate, using GraphPad Prism 7 (GraphPad Software, San Diego, CA, USA). Linear regression was conducted using GraphPad Prism 7 (GraphPad Software, San Diego, CA, USA). Analysis of species identifications was stratified to *A. castellanii* vs. other species to facilitate statistical analysis of demographic data including age and sex. All *p*-values were tested at α = 0.05 level of significance. 

## Figures and Tables

**Figure 1 pathogens-11-00661-f001:**
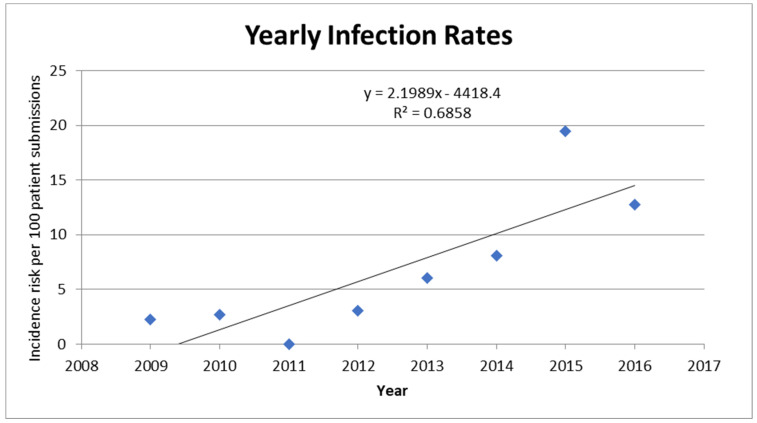
Proportion of samples sent for testing that were diagnosed as positive for *Acanthamoeba* per year from June 2009 to June 2016. Numbers plotted correspond to those found in Table 2. A linear curve was fitted to the graph (equation and R squared provided).

**Figure 2 pathogens-11-00661-f002:**
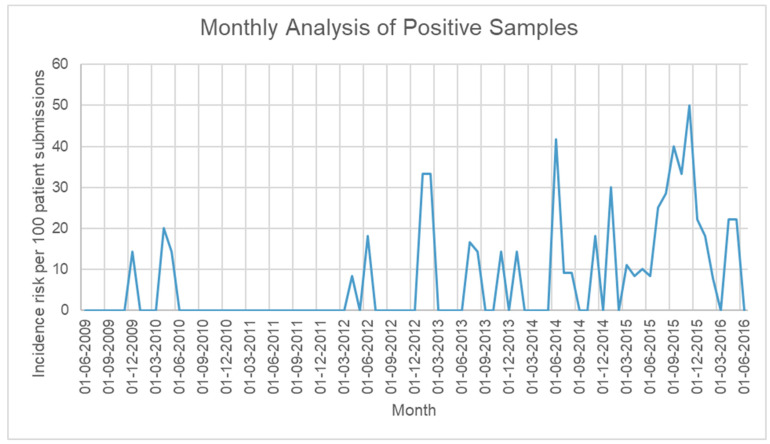
Proportion of samples sent for testing that were diagnosed as positive for Acanthamoeba per month from June 2009 to June 2016.

**Figure 3 pathogens-11-00661-f003:**
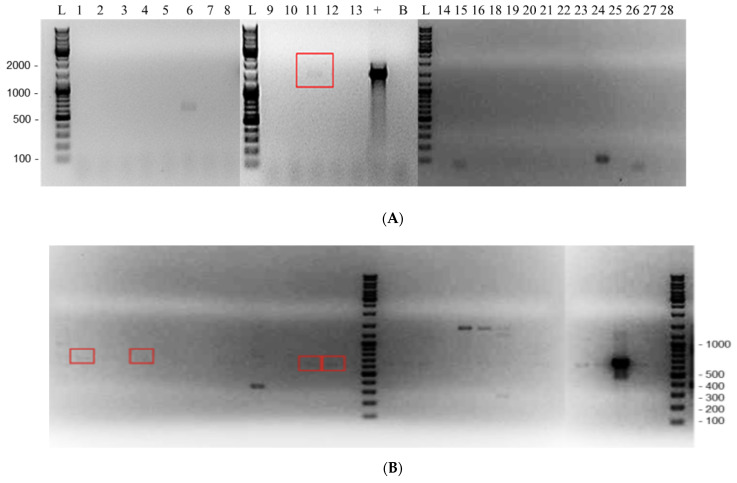
Agarose gel electrophoresis showing positive detection of various endosymbionts in 27 clinical Acanthamoeba samples. (**A**) Chlamydiales PCR. (**B**) Legionellales PCR. (**C**) Rickettsiales PCR. L—DNA ladder with base pairs indicated; 1 to 28—clinical samples; “+”—positive control of the respective PCR; “L+”—Legionellales-positive control; “C+”—*Coxiella*-positive control; B—negative control. Endosymbiont positives in clinical samples are indicated by red boxes. There was insufficient DNA in clinical sample # 17 to be analyzed, and sample 4 was subsequently removed from the study.

**Figure 4 pathogens-11-00661-f004:**
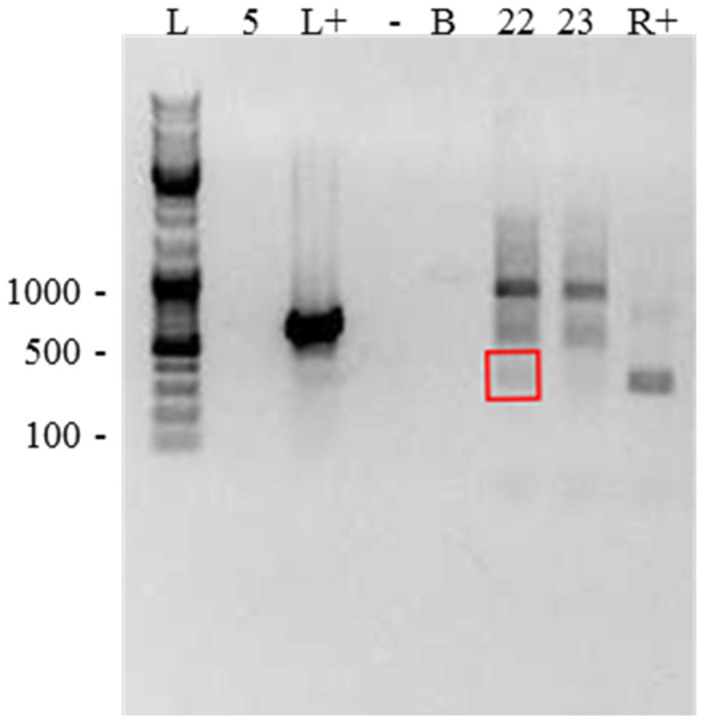
Gel showing DNA detected outside of cells, when sample DNA was extracted without a freeze–thaw step, to determine whether bacteria detected were present within the collected sample but outside of *Acanthamoeba* cells. Lanes as follows: L—DNA ladder; 5—sample #A5 (no DNA detected); L+—*Legionella*-positive control (strong band detected); -—negative control (no DNA detected); B—water (no DNA detected); 22—A22 (DNA of interest highlighted in red box); 23—A23 (no DNA of interest detected); R+—*Rickettsia* control (band detected).

**Table 1 pathogens-11-00661-t001:** Patient and sample baseline characteristics.

Number of samples tested for *Acanthamoeba*, N = 739 Positive: 86 (11.6%) Negative: 653 (88.4%)
Number of individuals tested for *Acanthamoeba*, N = 642 Positive: 50 (7.8%) Negative: 592 (92.2%)
Sex of individuals tested for *Acanthamoeba* Female: 332 (51.7%) Male: 306 (47.6%) Unknown: 4 (0.6%)
Age (years) of individuals tested for *Acanthamoeba* Mean: 50.4 +/− 22.74 Median: 51 (4 d–94 yr)
Sample type collected for *Acanthamoeba* testing Corneal Scrapings: 676 (91.5%) Contact Lens Solutions: 47 (6.4%) Culture Plate: 1 (0.1%) Unknown: 15 (2%)

**Table 2 pathogens-11-00661-t002:** Epidemiological data per year and proportion of positive tests.

	Total Tests Requested (# of Patients)	Tests Positive (# of Patients)	Infection Rate Per 100 Patient Submissions
2009 (starting July)	45	1	2.2
2010	75	2	2.7
2011	49	0	0
2012	98	3	3.1
2013	83	5	6.2
2014	124	10	8.1
2015	113	22	19.5
2016 (ending July)	55	7	12.7

**Table 3 pathogens-11-00661-t003:** Relationship of sex and age with AK positivity.

Variable	AK Positive	AK Negative	% Positive	Effect Measure	95% CI	*p*-Value
Sex Male Female	19 30	287 302	6.2% 9.0%	Odds Ratio 0.66	0.36 to 1.19	0.1803
Age (mean, median, and range)	Mean: 44.5 +/− 20.26 Median: 46 Range: 4 years–92 years	Mean: 51.3 +/− 22.97 Median: 53 Range: 4 days–94 years	N/A	Mean Difference 6.63	0.051 to 13.22	0.0482

**Table 4 pathogens-11-00661-t004:** Relationship of sex and age with *Acanthamoeba* species.

Variable	*A. castellanii*	Other Species	% *A. castellanii*	Effect Measure	95% CI	*p*-Value
Sex Male Female	1 3	5 3	16.7% 50.0%	Odds Ratio 0.2	0.013 to 2.18	0.5455
Age (mean, median, range)	Mean:39.5 +/− 24.09 Median: 38.5 years Range: 12 years–69 years	Mean: 48.6 +/− 15.27 Median: 47 years Range 23 years–70 years	N/A	Mean Difference 9.13	−15.93 to 34.18	0.4360

**Table 5 pathogens-11-00661-t005:** Relationship of sex and age with endosymbiont detection.

Variable	Endosymbiont	No Endosymbiont	Endo	Effect Measure	95% CI	*p*-Value
Sex Male Female	4 4	5 12	44.4%% 25.0%	Odds Ratio 2.4	0.49 to 12.63	0.3942
Age (mean, median, range)	Mean: 42.1 +/− 24.70 Median: 46 years Range: 4 years–69 years	Mean: 45.9 +/− 22.54 Median: 46 years, Range: 2 years–80 years	N/A	Mean Difference 3.82	−16.78 to 24.41	0.7050

**Table 6 pathogens-11-00661-t006:** Comparison of all 27 samples screened for endosymbionts and associated results.

Sample	Sex	Age	Sample Type	*Acanthamoeba* sp.	Endosymbiont Genus and/or Species
A1 *	F	31	Unknown	*A. castellanii*	*Rickettsia*, *Pseudomonas genticulata*
A2	F	61	Corn. Scrap.		*Legionella*
A3	F	63	Corn. Scrap.		
A5 *^	M	69	Corn. Scrap.		*Legionella*
A6 *	F	46	Corn. Scrap.	*A. hatchetti*	
A7	F	2	Unknown		
A8	F	46	Corn. Scrap.	*A. castellanii*	
A9 *	F	29	Corn. Scrap.		
A10 *	F	80	Corn. Scrap.		
A12 *	M	69	Corn. Scrap.	*A. castellanii*	
A13	M	69	Corn. Scrap.	*A. quina*	
A11 *	M	68	Contact lens		*Neochlamydiae hartmanellae*
A15	M	48	Corn. Scrap.	*A. polyphaga*	*Legionella*
A16 *	M	70	Corn. Scrap.		
A18*	M	70	Corn. Scrap.	*A. lenticulata*	
A14*	F	4	Corn. Scrap.		*Legionella, Bacillus*
A19 *	M	40	Corn. Scrap.	*A. griffini*	
A21 *	F	12	Corn. Scrap.		
A22 *^	F	12	Contact lens	*A. castellanii*	*Rickettsia*
A23 *^	F	12	Contact lens	*A. castellanii*	*Rickettsia*
A24 *	F	38	Corn. Scrap.		
A25	F	49	Plate	*A. lenticulata*	
A26 *	F	50	Corn. Scrap.		
A28	F	23	Corn. Scrap.	*A. hatchetti*	
A27 *	M	44	Corn. Scrap.	*A. polyphaga*	*Legionella*
A20 *	F	25	Corn. Scrap.		

(*) indicates a sample analyzed by 16S PCR. (^) indicates a sample used for confirming endosymbiosis. Corn. Scrap. indicates a sample coming from a corneal scraping. Samples A22 and A23 were obtained from the same patient on the same date.

**Table 7 pathogens-11-00661-t007:** Summary of T genotypes and corresponding putative species [13].

Genotype	Associated Species
T1	*A. castellanii*
T2	*A. palestinensis* *A. polyphaga* Unknown species
T3	*A. griffini*
T4	*A. castellanii* *A. rhysodes* *A. polyphaga* *A. triangularis* Unknown species
T5	*A. lenticulata*
T6	*A. palestinensis*
T7	*A. astronyxis*
T8	*A. tubiashi*
T9	*A. comandoni*
T10	*A. culbertsoni*
T11	*A. hatchetti*
T12	*A. healyi*
T13	Unknown species
T14	Unknown species
T15	*A. jacobsi*
T16	Unknown species
T17	Unknown species
T18	*A. tubiashi*
T19	Unknown species
T20	Unknown species
T21	*A. pyriformis*
T22	*A. royreba*
T23	*A. bangkokensis*

## Data Availability

All available data are included in the manuscript.

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
