# Peer review of "Surveillance of Amoebic Keratitis-Causing Acanthamoebae for Potential Bacterial Endosymbionts in Ontario, Canada"

_pathogens, 2022, doi:10.3390/pathogens11060661_

Round 1

Reviewer 1 Report

This is a report of findings that are consistent with the fact that there is a small amount of endosymbionts present in the AK samples. They are not dependent on sex but may correlate with age. It appears to be isolated and not very strong since the cases are few in number.. Adding more samples is warranted.

The conclusions are consistent with the evidence presented but the
significance as regards treatment are not forthcoming. The somewhat
address the main question but this could be improved.

Reviewer 2 Report

This manuscript reports the experiences of a Canadian Provincial clinical parasitology lab monitoring Acanthamoeba Keratitis (AK) from 2009 until 2016. These cases have been correlated with bacterial endosymbionts.  This is of course represents a huge amount of work and would be met with interest in the AK field and more generally in the Free-Living amoebae field. This work also provides evidence for an increase in the frequency of infections in this sample area rather than an increase in detection or reporting.

In my opinion, there are two major issues to be addressed, however.  Despite the argument made by the authors in the discussion, the co-isolated bacteria cannot confidently be considered as endosymbionts (although many are likely to be) as the authors would have to show that these bacteria are actually carried and growing inside the Acanthamoeba. From the methods section (Lines 413-419) it is evident that this has not been carried out rigorously enough. Even if this method was sufficient to differentiate contaminating bacteria from genuine endosymbionts (which I doubt) it would have to be done for all samples not just a subset.  Even if the freeze thaw only liberated bacteria from the cysts/trophozoites which were not detected in the supernatant, the status of identified bacteria as symbionts versus amoeba resistant pathogenic bacteria would have to be determined experimentally. Many of the bacteria detected/identified have indeed been shown to be endosymbionts of Acanthamoeba but it is not safe to assume that they are playing the same role in these isolates with these particular Acanthamoeba strains. Have the authors isolated sequences using the clinical samples that would contain the expected ocular contaminants (Staphylococcus etc)? The discussion states that “This procedure was done on 3 of the 9 samples …”(Line 290) but it is not clear what procedure was in these cases.

I suggest changing the title of the paper to reflect this considerable uncertainty or commit to culture experiments if these samples are still available and have remained viable. The title may be modified by placing the word “potentially” before “endosymbionts” and including an explanation in the text, ideally also in the abstract.

The second issue relates to the classification of Acanthamoeba. The classification of Acanthamoeba is not resolved and although some species names have been used in the literature, these are largely meaningless so that for example there is no difference between Acanthamoeba castellanii and Acanthamoeba polypgaga. Acanthamoeba is classified into some 23 genotypes (named T1 to T22) based on 18S sequences, each of which equates or approximates to a species, and isolates which group with A. castellanii and A. polyphaga both fall within T4 (see Putaporntip et al, 2021 and the references therein). This being the case it would be much more productive and useful to use the T type classification instead of the very imperfect species names.  T4 is the most commonly isolated genotype and the most commonly associated with pathogenicity when its relative abundance is considered (Maciver et al, 2013 see at the end of this report).

Table 1 contains the same information as in the text (Lines 71 – 80) but it needs more information to be understood more readily.  So for example “Number of Samples, N=739” would be better as “Number of Samples tested for Acanthamoeba, N=739” and so on for the other rows in table 1

Some typos

Line 75 Remove “(4/642)” since as this is a repeat.

Line 102, double full stop; Line103 space between “Table” and “3a” needed.

Table 3a format so that “Odd Ratio” is in the correct place.

Line 196 Legend to Figure 3. Ensure that the well description in the legend matches the well labels in the figure

Line 255 The difference between the two quoted % prevalence is not valid because of the low numbers here?  Also the primers used here would not pick up all possible bacteria?

Line 406 Capitals are needed for both legionella and chlamydia here.

References. It is not necessary to state when references were read?  This has been done for a number of the cited references. Refs 1,2, 9, 12 etc

Some information missing from reference 11. The full ref is:-

Carnt, N., Hoffman, J. J., Verma, S., Hau, S., Radford, C. F., Minassian, D. C., & Dart, J. K. (2018). Acanthamoeba keratitis: confirmation of the UK outbreak and a prospective case-control study identifying contributing risk factors. British Journal of Ophthalmology102(12), 1621-1628.

Maciver, S. K., Asif, M., Simmen, M. W., & Lorenzo-Morales, J. (2013). A systematic analysis of Acanthamoeba genotype frequency correlated with source and pathogenicity: T4 is confirmed as a pathogen-rich genotype. European journal of protistology49(2), 217-221.

Putaporntip, C., Kuamsab, N., Nuprasert, W., Rojrung, R., Pattanawong, U., Tia, T., ... & Jongwutiwes, S. (2021). Analysis of Acanthamoeba genotypes from public freshwater sources in Thailand reveals a new genotype, T23 Acanthamoeba bangkokensis sp. nov. Scientific reports11(1), 1-13.

Round 2

Reviewer 1 Report

The manuscript is improved sufficiently.

Reviewer 2 Report

The authors have addressed the issues that I raised adequately